# Accuracy of 3D printed scan bodies for dental implants using two additive manufacturing systems: An in vitro study

Liam J. Hopfensperger[1], Georgi Talmazov[2], Rami Ammoun[3], Christian Brenes[4], Sompop Bencharit[1,4,5]*

**1** Department of Oral & Craniofacial Molecular Biology, Philips Institute for Oral Health Research, School of Dentistry, Virginia Commonwealth University, Richmond, Virginia, United States of America, **2** Private Practice, Washington DC, United States of America, **3** Department of Prosthodontics, School of Dentistry, Virginia Commonwealth University, Richmond, Virginia, United States of America, **4** Department of Oral Rehabilitation, College of Dental Medicine, Medical University of South Carolina, Charleston, South Carolina, United States of America, **5** Department of Biomedical Engineering, College of Engineering, Virginia Commonwealth University, Richmond, Virginia, United States of America

* benchari@musc.edu

**Data Availability Statement:** All relevant data are within the paper.

**Funding:** The authors received no specific funding for this work.

## Abstract

This study compared the accuracy of implant scan bodies printed using stereolithography (SLA) and digital light processing (DLP) technologies to the control (manufacturer's scan body) Scan bodies were printed using SLA (n = 10) and DLP (n = 10) methods. Ten manufacturer's scan bodies were used as control. The scan body was placed onto a simulated 3D printed cast with a single implant placed. An implant fixture mount was used as standard. The implant positions were scanned using a laboratory scanner with the fixture mounts, manufacturer's scan bodies, and the printed scan bodies. The scans of each scan body was then superimposed onto the referenced fixture mount. The 3D angulation and linear deviations were measured. The angulation and linear deviations were 1.24±0.22˚ and 0.20±0.05 mm; 2.63±0.82˚ and 0.34±0.11 mm; 1.79±0.19˚ and 0.32±0.03 mm; for the control, SLA, and DLP, respectively. There were statistical differences (ANOVA) among the three groups in the angular (p<0.01) or linear deviations (p<0.01). Box plotting, 95% confidence interval and F-test suggested the higher variations of precision in the SLA group compared to DLP and control groups. Scan bodies printed in-office have lower accuracy compared to the manufacturer's scan bodies. The current technology for 3D printing of implant scan bodies needs trueness and precision improvements.

## Introduction

Advancements in digital technologies allow an impression of a dental implant using a scan body. Implant digital impressions and workflows have been shown to be a preferred method for both patients and clinicians [1–3]. Digital impressions can produce a CAD/CAM dental cast with accuracy similar to that of conventional impression and gypsum dental casts [4, 5]. Digital implant impressions have shown to save cost and be more effective for a single implant

**Competing interests:** Author Sompop Bencharit is a paid consultant and speaker for Formlabs and ZimVie, outside the submitted work. Author Christian Brenes is also a paid consultant and speaker for Sprinkray, outside the submitted work. The manufacturers did not provide any support to this project and did not play any role in the experimental design, result interpretation, manuscript preparation, or the submission of this work. There are no patents, products in development or marketed products associated with this research to declare. This does not alter our adherence to PLOS ONE policies on sharing data and materials.

restoration [6, 7]. Recent advances in desktop 3D printing technology have changed the practice of dentistry. Together with the advancements of prosthetic design software, clinicians now are able to design and fabricate prosthetic devices and appliances in their office [8, 9]. Stereolithography (SLA) and digital light processing (DLP) are perhaps the most common in-office 3D printers used nowadays [10, 11].

When a single implant requires a digital impression, clinicians usually use either an implant's manufacturer specific implant scan body or a third party implant scan body specific to a certain implant fixture/system. The implant scan body is usually made of polymer materials such as polymethyl methacrylate (PMMA) or polyether ether ketone (PEEK), or titanium alloy metal or other materials [12]. An accuracy of a digitized single implant position transfer to a 3D printed cast or digital implant cast using PEEK scanbody of a single implant PEEK scanbody is shown on average ~105–127 μm with 0.22˚-1.25˚ for an *in vitro* study [13]. Similarly, an average range of ~36–57 μm was shown in an in vitro study for a single implant digital impression using a scan body-healing abutment combination system [14]. However, a systematic review suggested that the degrees of accuracy can be varied based on different studies for multiple implant digital impressions [15]. The older studies demonstrated highest deviations of on average of 1 mm or 1,000 μm [16] Most later studies demonstrated the accuracy of <85 μm [15, 17–20]. However, some recent studies demonstrated that the accuracy can be >160 μm [21, 22].

Implant dentistry has evolved over the last 5 decades with numerous implant-abutment connections and designs. This can present challenges for clinicians in making digital impressions that require specific implant scan bodies. This study therefore presented a proof of principle by applying two most common 3D printing technologies, SLA and DLP, to print a scan body and comparing the positions of the printed scan body to the conventional manufacturer's provided scan body. The hypothesis was that there was no statistically significant difference between the three-dimensional angulation and linear deviations between the two printed types of scan bodies and the control manufacturer's implant scan bodies.

## Materials and methods

### Implant scan body design

The external dimensions of a manufacturer-produced scan body (PSA3SCAN Scan Body, ZimVie) were measured using a digital caliper. FreeCAD was used to reverse engineer an STL file of the manufacturer produced scan body due to its ability to produce basic geometry with exact measurements (Fig 1). The idea was to create a scan body mimicking a situation when a commercial scan body is not available.

### 3D printing and post-processing methods

Ten scan bodies (Fig 2) were printed using an SLA printer (Form 3B, Formlabs) and ten scan bodies were printed using a DLP printer (Sprinray Pro95, Sprinray). The sample size (n = 10) for each group was determined using a similar previous study [14]. For the Formlabs scan bodies, the scan body was exported into a 3D CAD softwares in STL format (Preform, Formlabs), where printing supports and rafts were added to the scan bodies. Several batches, 10 scan bodies each, were printed in resin (Model 2 Resin, Formlabs) at a resolution of 50 μm (Form 3B, Formlabs) to verify the general accuracy of the scan body dimensions. Each batch, including the final batch used for analysis, had print supports removed and was post-processed according to manufacturer recommendations (Form Wash and Form Cure, Formlabs). The Form Cure cycle was set to 30 minutes at 60˚C, following manufacturer recommendations.

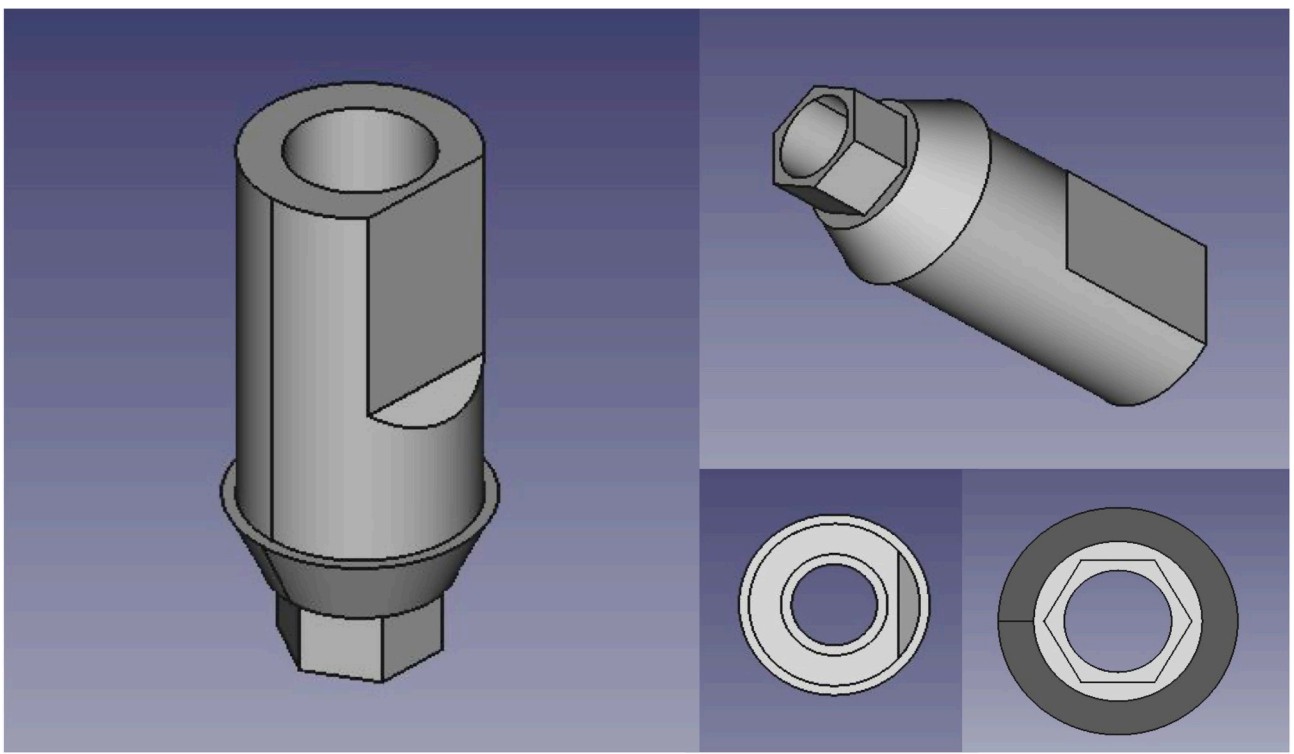

**Fig 1. Scan body design.**

For the SprintRay scan bodies, the same STL file which produced the final batch of Formlabs scan bodies was imported into their proprietary 3D CAD software (RayWare, SprintRay), where printing supports and rafts were added to the scan bodies. One batch of 10 scan

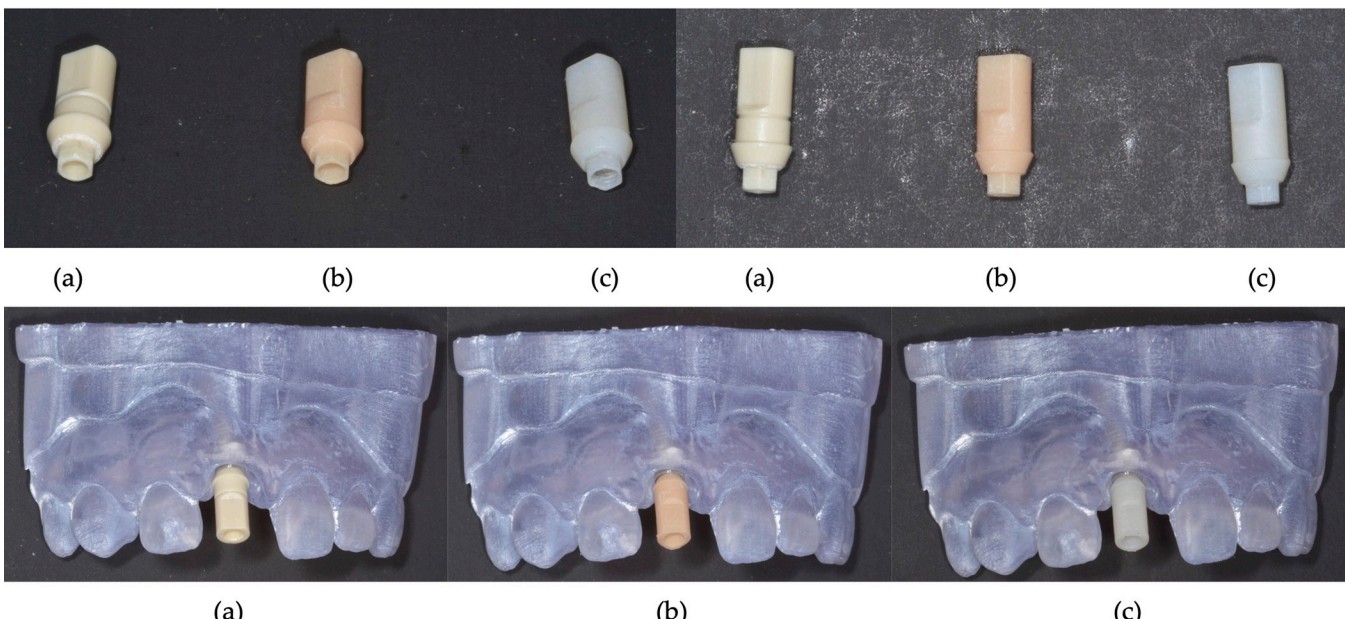

**Fig 2.** Scan bodies: (a) manufacturer's scan body; (b) SLA printed scan body; and (c) DLP printed scan body, both lone standing and in dental cast.

bodies was printed in resin (Crown and Bridge Resin (BL1), DENTCA) at a resolution of 95 μm (SprintRay Pro95, SprintRay). Each scan body had print supports removed and was post-processed. This involved a 3 minute wash in 91% isopropyl alcohol (CLEANI, Ackuretta) and manufacturer-recommended curing settings (5 minutes, ProCure 2, SprintRay).

### Dental implant cast fabrication

A dental cast (Fig 3) with a dental implant was fabricated using the process described in Adams et al. [23]. Briefly, the original dental cast and implant planning were obtained from an unidentified patient who was missing a maxillary right central incisor from the implant clinic database. The Virginia Commonwealth University Office of Research and Innovation reviewed and approved the study protocol. The use of unidentified data was approved without requiring patient consent (IRB no. HM20009486). The CBCT scans, i-CAT FLX V10 (Kavo Dental, Brea, CA) with standard implant scan parameters (16 cm in depth, 10 cm in height, 0.3-mm voxel size, 8.9-second scan time, 3.7-second exposure time, 120 kVP, 5 mA, and 501.3 mGy/cm2) and the intraoral scans (TriOS 3; a3Shape A/S, Copenhagen, Denmark) were included in the implant treatment planning using the Implant Studio 2020 (3Shape A/S) to replace the missing tooth with a Tapered Screw Vent (TSV) (3.7mm x 13mm) dental implant (ZimVie). The dental cast was printed using Dental Clear LT resin V2 (Formlabs) to allow placement of a dental implant without cracking or fracturing of the cast [23]. A surgical guide was fabricated using Surgical Guide Resin (Formlabs) at a resolution of 0.05 mm. The dental casts and surgical guides were post-processed according to the manufacturer's

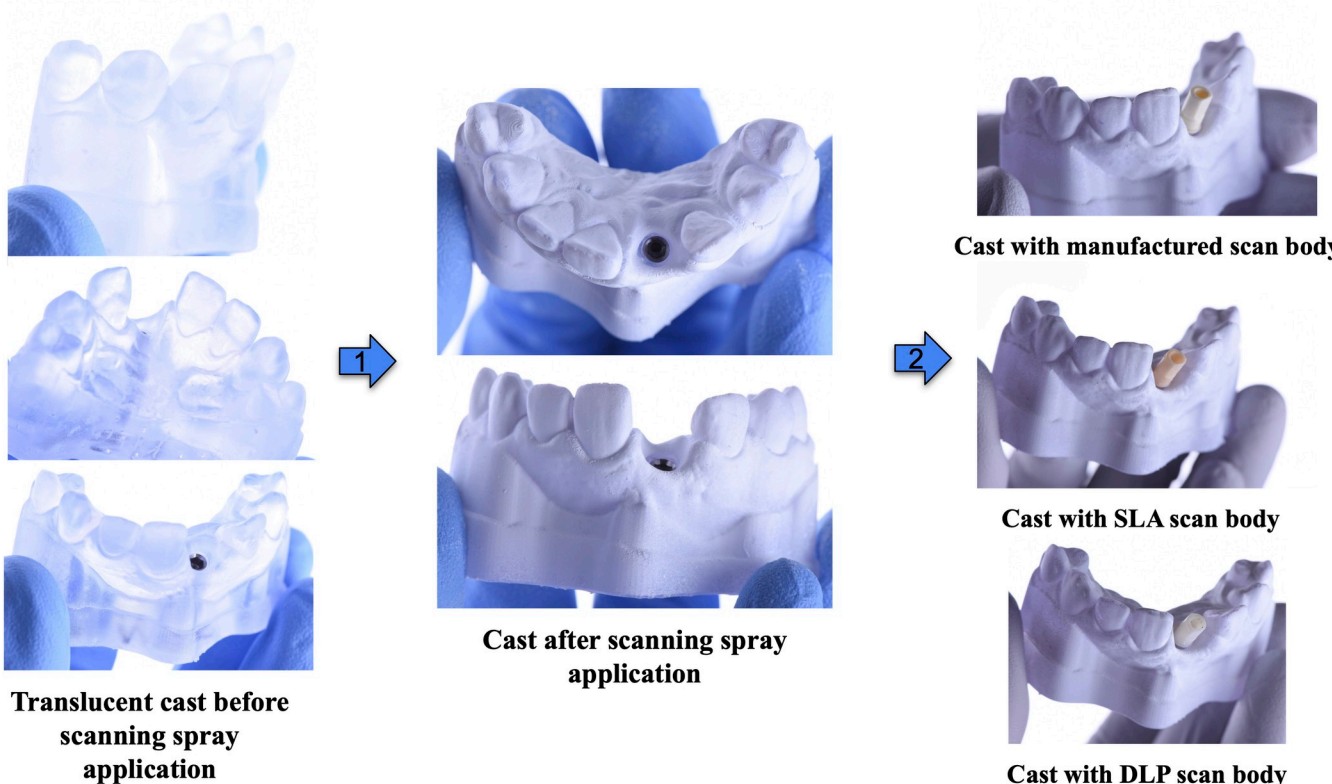

**Translucent cast before scanning spray application**

**Cast after scanning spray application**

**Cast with manufactured scan body**

**Cast with SLA scan body**

**Cast with DLP scan body**

**Fig 3. Simulated dental cast with implants and scan bodies in place.**

recommendations (Form Wash & Form Cure, Formlabs). The implant was placed according to the implant guided surgical protocol [23].

## Scan body position acquisition and comparison

The study workflow is shown in Fig 4. An implant cover screw was placed onto the implant to protect the internal implant connection. The cast was then sprayed with a CAD/CAM spray to cover the reflective surface (Scanning Spray, Henry Schein) and facilitate the scanning. Ten sets of scan bodies, each containing one of each type of scan body and the same shared screw (to minimize screw variations), were numbered. The cast was scanned with each of the three scan bodies of each set using a desktop scanner (E4 Generation RED, 3Shape) and its corresponding software system (Dental System, 3Shape). The cast was also scanned with a dental implant fixture mount (3.5mmD Fixture Mount/Transfer FMT3, ZimVie) to act as a standard reference for all three groups of scan bodies. To minimize the effects of scanning spray to the dimension of the specimen, the spray was performed one time prior to scanning of the fixture mount, and all scan bodies. There was no application of the spray onto the fixture mounts or the scan bodies. All scanning procedures for all four samples (fixture mount, manufacturer's scan body, SLA scan body, and DLP scan body) in each cast were done consecutively without additional scanning spray application.

Additionally, each of the four implant attachments (fixture mount, 3 scan body types) was mounted into a lone-standing Zimmer implant of the same model as in the cast. The implant

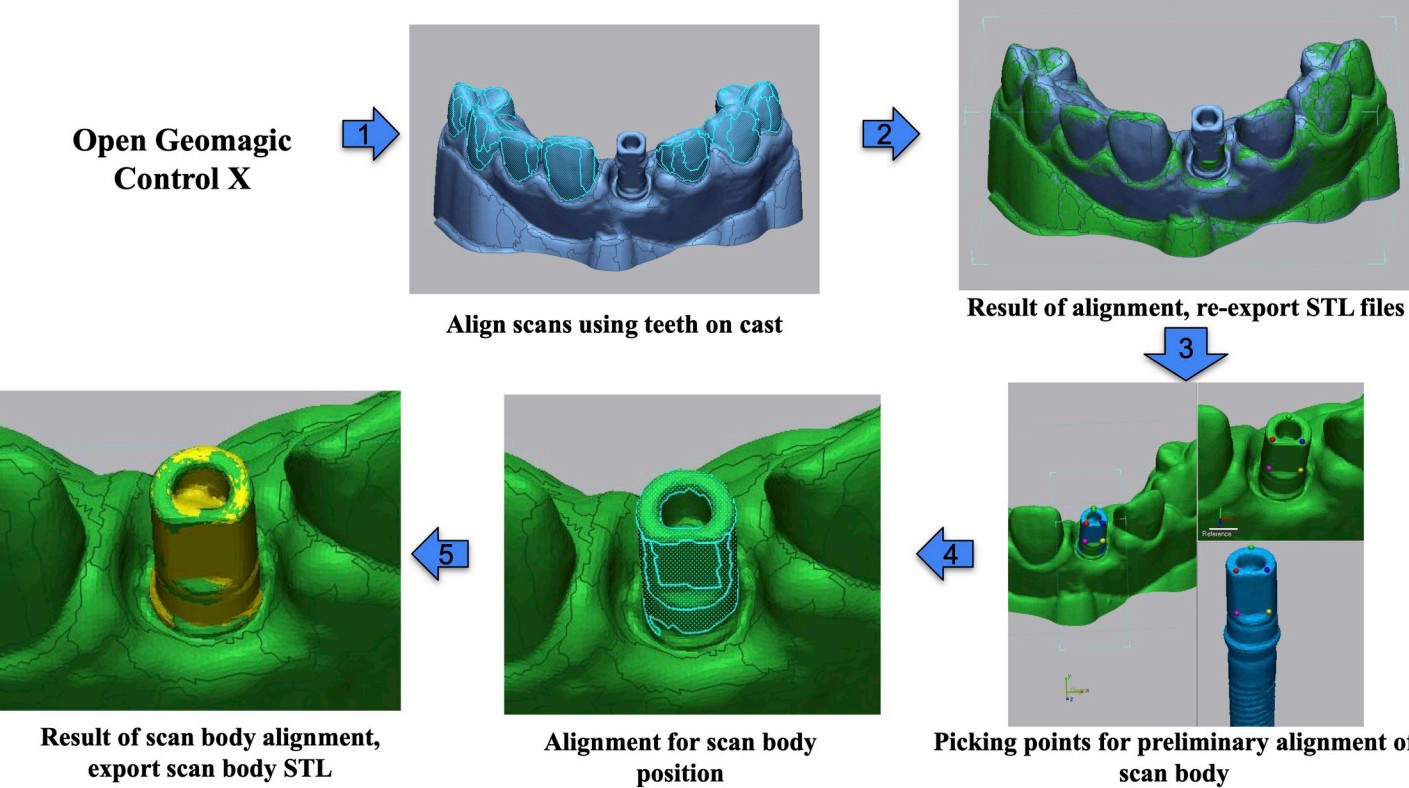

**Open Geomagic Control X**

**Align scans using teeth on cast**

**Result of alignment, re-export STL files**

**Result of scan body alignment, export scan body STL**

**Alignment for scan body position**

**Picking points for preliminary alignment of scan body**

**Fig 4. Superimposition of the casts and the scan bodies.** Once the scans of the cast were aligned using the remaining teeth, the flat part of the scan body with four reference points and the flat side of the top of the scan body were then used to define the positioning of the screw access. The central axis of the screw access was then used to compare the linear and angular deviations.

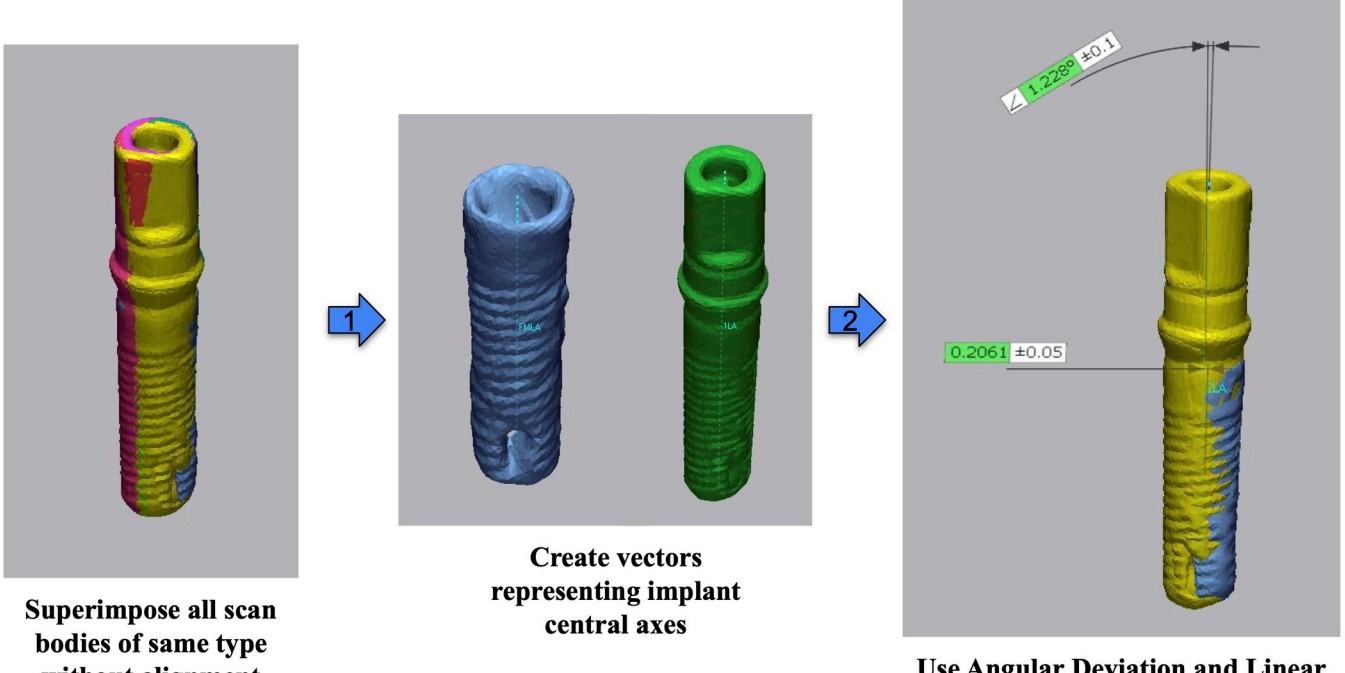

**Fig 5. Superimposition of the scan bodies and measurements.**

itself was scanned multiple times from various angles using the same desktop scanner (E4, 3Shape), as were each of the implant-attachment complexes. These scans were aligned using Geomagic Control X (2022.1, 3D Systems) and the resulting spliced models were cleaned using Meshmixer (2017, Autodesk). All scans were exported as STL files and superimposed using Geomagic Control X (2022.1, 3D Systems) as follows (Fig 5).

For each type of scan body, acquired data was compared to the fixture mount-cast complex using best-fit alignment of only the cast's teeth, excluding the scan bodies and soft tissue areas. These post-alignment models were then exported as STL files with new position and orientation information. The post-alignment models were then used as reference to compare the implant-attachment complex scans. This was completed by selective best-fit alignment which only included the supragingival outer scan body surfaces and excluding areas inside of the screw channel. The post-best fit complex was then exported as an STL file with new position and orientation information. Finally, all post-alignment scan body-implant analog complex models were reimported into a new Geomagic Control X (2022.1, 3D Systems) trial set project. At this point, the central axis was determined using a software algorithm and made into a vector. In the case of the implant-fixture mount complex, this complex was not accurately cylindrical due to the shape of the fixture mount. Thus, selective best-fit alignment was again performed to align the implant scan to the implant-fixture mount complex. This post-best fit implant scan was then used to generate the reference vector. The trueness and precision evaluations in this study were determined by the linear and angular deviations of the long axis of the scan body compared to the long axis of the fixture mount instead of standard methods such RMS deviation to be reflective of clinical practice using similar methods as previous implant guided surgery studies and to eliminate minor variations possible from the reengineering of the scan body [23–25]. Deviations in angular and linear position were calculated by

comparison of all other vectors within the set to this reference vector. The aforementioned process was completed for all three types of scan bodies and the data gathered for statistical analysis.

## Statistical analysis

Similar statistical analyses from previous studies were utilized [23–25]. The means and standard deviations were calculated for each group. 95% confidence intervals were used to assess the range of angulation and linear deviations. Box plotting was used to demonstrate the overall positioning deviation of the implant scan bodies. To determine the trueness, single factor multivariate analysis (ANOVA) with $\alpha$ = 0.05 was used to determine differences among the three groups. t-test analysis with unequal variants with = 0.05 was used to determine the differences between each pair. To determine the precision, one-tail F-test with $\alpha$ = 0.05 was used to determine whether any statistically significant differences in the variance between each pair.

## Results

The summary of the data and statistical analyses were presented in Table 1 (See also the Box Plottings in Figs 6 and 7). The angulation deviation of the control (manufacturer's scan bodies) with 95% CI [1.08˚, 1.40˚] was the best, followed by the DLP printed ones with 95% CI [1.64˚, 1.94˚], and the SLA printed ones with 95% CI [2.00˚, 3.26˚]. There was a statistical difference among the three groups in terms of trueness of the angulation deviations when comparing the mean angular deviation (ANOVA, p<0.01). t-tests also showed significant differences in all comparing groups. In terms of precision or consistency of the deviations, there were statistically significant differences between the control and SLA groups, as well as the DLP and SLA group (F-test, p<0.01). However, there was no significant difference seen between the DLP and control groups (F-test, p = 0.34). The SLA group clearly had lower trueness and precision in the angulation dimensions compared to the other group. The manufacturer group was best overall in the angulation dimension. The DLP group had similar precision as the control group.

The linear deviations (mm) were 95% CI [0.17, 0.24], 95% CI [0.26, 0.42], and 95% CI [0.30, 0.35] for the manufacturer, SLA, and DLP groups, respectively. In terms of trueness,

**Table 1. Data and statistical analyses for angulation and linear deviations.**

| | Scan body type | Mean | SD | Range | Min | Q1 | Q3 | Max | 95% CI | ANOVA |
|---|---|---|---|---|---|---|---|---|---|---|
| **Angulation (˚)** | Manufacturer | 1.24 | 0.22 | 0.61 | 0.97 | 1.06 | 1.42 | 1.58 | [1.08, 1.40] | <0.01 |
| | SLA | 2.63 | 0.05 | 2.70 | 1.29 | 2.19 | 3.06 | 3.99 | [2.00, 3.26] | |
| | DLP | 1.79 | 0.82 | 0.59 | 1.45 | 1.68 | 1.94 | 2.05 | [1.64, 1.94] | |
| **Linear (mm)** | Manufacturer | 0.20 | 0.05 | 0.15 | 0.14 | 0.16 | 0.23 | 0.30 | [0.17, 0.24] | <0.01 |
| | SLA | 0.34 | 0.11 | 0.37 | 0.15 | 0.27 | 0.41 | 0.52 | [0.26, 0.42] | |
| | DLP | 0.32 | 0.03 | 0.11 | 0.27 | 0.31 | 0.34 | 0.38 | [0.30, 0.35] | |
| | Scan body type | Compared to | t-test* | F-test* | | | | | | |
| **Angulation (˚)** | Manufacturer | SLA | <0.01 | <0.01 | | | | | | |
| | Manufacturer | DLP | <0.01 | 0.34 | | | | | | |
| | SLA | DLP | 0.01 | <0.01 | | | | | | |
| **Linear (mm)** | Manufacturer | SLA | <0.01 | 0.02 | | | | | | |
| | Manufacturer | DLP | <0.01 | 0.06 | | | | | | |
| | SLA | DLP | 0.65 | <0.01 | | | | | | |

*$\alpha$ = 0.05

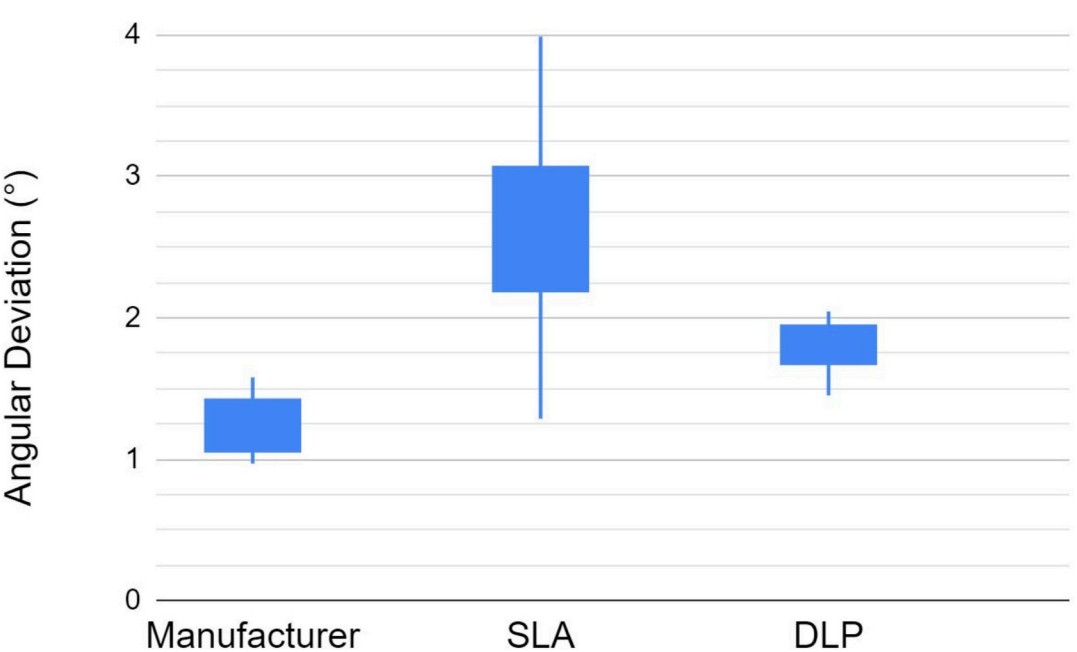

**Fig 6. Angulation deviations (˚) of the manufacturer's scan bodies (control), the SLA and DLP printed scan bodies.**

there were statistical significant differences found among the means of the three groups, ANOVA, p<0.01). There were statistically significant differences between the manufacturer group and the two test printed groups (t-test, p<0.01). However, there was no statistically significant difference between the SLA and DLP groups. This suggests that the linear trueness of the manufactured scan bodies was better than the two printed ones. In terms of linear

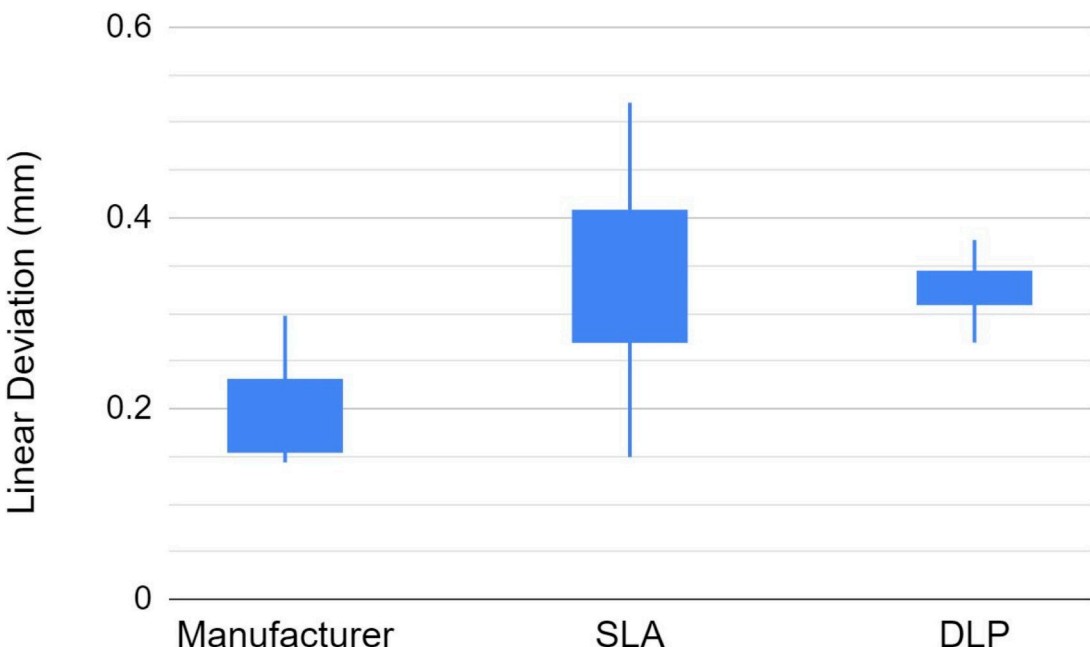

**Fig 7. Linear deviations (mm) of the manufacturer's scan bodies (control), the SLA and DLP printed scan bodies.**

precision, there were statistically significant differences between the manufacturer and the SLA groups (F-test, p = 0.02) and the SLA and DLP groups (F-test, p<0.01). However, there was no statistically significant difference between the manufacturer and the DLP group (F-test, p = 0.06). This suggests that the DLP printed scan bodies while having lower linear trueness, the DLP printed linear trueness was similar to the manufacturer's ones. In any case, the manufacturer's scan bodies demonstrated overall best trueness and precision in the linear dimension.

## Discussion

The study was one of the first to demonstrate the accuracy of 3D printed implant scan bodies compared to the manufacturer's scan bodies as a control. The results demonstrated clearly that the control scan bodies have the best trueness and precision in both angulation deviations and linear deviations with 95% CI [1.08˚, 1.40˚] and 95% CI [0.17 mm, 0.24 mm], respectively. The average angulation and linear deviations of the manufacturer group were 1.24˚ and 0.2 mm, respective. These values were significantly lower than the two printed scan body angulation and linear deviations in this study were on average 2.63˚ and 0.34 mm for SLA, and 1.79˚ and 0.33 mm for DLP groups. While the deviations of the manufacturer's scan bodies were slightly in the high-end of the spectrum, the deviations of SLA and DLP printed scan bodies were clearly too high. [12]. While the printed scan bodies need to have lower deviation values to be clinically acceptable for a single implant, [15, 17–20] this study provides the first proof of principle that if a scan body is not available it may be possible to design and create a scan body *in silico* and then print it using an in-office 3D printer. In terms of the two technologies, SLA or DLP, both technologies have clearly lower trueness than the control. In terms of precision, however, it appears that the DLP technology demonstrates similar precision with the manufacturer's scan bodies. This suggests DLP 3D printed scan bodies provide better consistency of printed scan bodies compared to the SLA. This variation or inconsistency may increase in the case of multiple implant impressions as well as complexity of digital designing, printing, and post-processing. In-office 3D printed technology needs to be improved before clinically acceptable for implant scan body fabrication. It is important to note that in this study, the scan body was reengineered and printed to simulate a situation when a manufacturer's scan body is not available such as when the implants are no longer produced by the manufacturer or the implants were placed in another country. The reengineered and printed scan bodies here were aimed to mimic this clinical situation. If the original STL file of the scan body was used, it is possible that the printed scan bodies may provide a closer trueness and precision values as the original scan bodies compared to the reengineered ones.

Dental implantology has evolved dramatically since PI Branemark introduced the concept of osseointegration in the 80s [26]. It was estimated that over 3 million dental implants are being placed in the US alone each year [27]. At the beginning of 21 century, there was a report of 25 implant manufacturers and 100 implant systems [28]. However, the numbers may have been out of date. More dental implants have been developed overseas as well as in the US. Dental and medical tourism creates a common situation when a patient has dental implants surgically placed overseas and then presents to a dental office with no known record of the implant used [29]. While on one hand, identification of an implant is a challenge [29], on the other hand, it can also be a challenge in finding a component for making an impression or a digital intraoral scan. This challenge can also be applicable to old implant systems that their parts and components may not be readily available. This issue soon can be a major problem due to the fact that patients are living longer and many of them have had previous implant treatments [30]. The applications of in-office manufacturing for dental appliances such as implant guides

has become popular in the last 6–7 years since the introduction of an inexpensive desktop 3D printer available for dental offices and dental laboratories [8, 9, 31]. Ability to manufacture an implant scan body would allow clinicians to be able to make digital impressions for any dental implants with appropriate design and manufacturing process similar to engineer manufacturing. This study provides perhaps the first indication of the potential of manufacturing implant scan bodies in-office. There is a learning curve of the applications of CAD design software such as FreeCAD [32]. Furthermore, since the scan body part is relatively small and requires a precise fitting into the implant fixture, the printing support design, printing angulation, and post-processing become very important in the production of consistently precise scan bodies, perhaps more so than implant surgical guides or dental casts [33, 34].

While in theory, SLA and DLP technologies may provide differences in the trueness and precision of the printed products, this study points out the need to improve trueness of both technologies but improve precision for SLA. A systematic review demonstrated similar ranges of trueness and precision between SLA and DLP printers [35]. A study comparing resin printed interim crowns demonstrating no statistical significance between the intaglios of the crowns printed by SLA and DLP printers [36]. SLA and DLP performed similarly in the fabrication of orthodontic retainers [37], implant surgical guides [38], or dental casts [39, 40]. Both SLA and DLP technologies are sensitive to changing in printing layer thickness. SLA appears to increase accuracy as the printing layer thickness decreases, while for the DLP, at least one study has suggested that the optimal printing layer for DLP is 50 μm [41]. A recent study suggested that SLA technology may provide better intaglio surface accuracy for denture base printing compared to DLP technology [42]. However, it is possible that other available 3D printing technologies may have different accuracy especially for printing implant scan bodies.

It is important to discuss the limitations of this study. First, this is an *in vitro* study, while the results demonstrate that similarity with the manufacturer's scan bodies, there is a need to validate the results *in vivo*. Second, the accuracy and consistency of the fabricated scan bodies is essential to the appropriate fitting of the scan body to the implant fixture. There may be a learning curve and technical difficulties when clinicians want to apply this technology into their practices. The size of the small screw access hole and the internal hex structure can be difficult to manipulate during printing and post-processing processes. Third, this study only compared two 3D printing technologies. The results therefore may not be generalizable to other types of 3D printers. The accuracy of particular 3D printers within the same technology category may also influence the results. This study utilized SpringRay for the DLP printer and Form 3B for SLA printer. The results, while providing general comparison, may be varied based on differences in particular printer's printing accuracy. Fourth, the study only examined a single implant situation with one implant-abutment connection system. The results are applicable to a situation of a single tooth implant in this particular internal hex connection. However, multiple implant situations or different implant-abutment connections may have different challenges. Finally, this study utilized the known dimension of the manufacturer's scan body. In real situations when a scan body is not available such as in the older implant systems or implants from other countries, the reverse-engineering process can be challenging and difficult to do. Additionally, variations from implant level or abutment level impression may also further complicate the design and fabrication process of the 3D printed scan body. Considering these limitations, future studies should include improving the design and printing trueness and precision prior to human clinical trials. Reverse-engineering of old dental implant parts as well as scan bodies will be an important topic as part of the future research. Future studies of multiple types of implant connection and platforms would also be an important next step.

## Conclusions

Manufacturer's implant scan bodies have better trueness compared to the SLA and DLP printed scan bodies in both angulation and linear deviations. The DLP technology provides comparable precision with the manufacturer's implant scan bodies but better than SLA technology. Current in-office 3D printed scan bodies need to be improved to be used clinically for implant impressions.

## Author Contributions

**Conceptualization:** Liam J. Hopfensperger, Sompop Bencharit.

**Formal analysis:** Liam J. Hopfensperger.

**Funding acquisition:** Sompop Bencharit.

**Investigation:** Liam J. Hopfensperger.

**Methodology:** Sompop Bencharit.

**Project administration:** Sompop Bencharit.

**Resources:** Rami Ammoun, Christian Brenes, Sompop Bencharit.

**Software:** Liam J. Hopfensperger, Georgi Talmazov.

**Supervision:** Rami Ammoun, Christian Brenes, Sompop Bencharit.

**Validation:** Liam J. Hopfensperger, Georgi Talmazov, Sompop Bencharit.

**Visualization:** Liam J. Hopfensperger.

**Writing – original draft:** Liam J. Hopfensperger, Sompop Bencharit.

**Writing – review & editing:** Liam J. Hopfensperger, Georgi Talmazov, Sompop Bencharit.

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
