## [Decision Letter · Decision Letter 0]

3 Jan 2023

PONE-D-22-29428Accuracy of 3D printed scan bodies for dental implants using two additive manufacturing systems: an in vitro StudyPLOS ONE

Dear Dr. Bencharit,

Thank you for submitting your manuscript to PLOS ONE. After careful consideration, we feel that it has merit but does not fully meet PLOS ONE’s publication criteria as it currently stands. Therefore, we invite you to submit a revised version of the manuscript that addresses the points raised during the review process.

Thank you for this interesting submission. Based on the reviewers' comments, the manuscript can't be accepted in the current form.  Hence, I invite you to address the concerns raised by the reviewers, make the necessary changes, and resubmit the revised manuscript.  

We look forward to receiving your revised manuscript.

Kind regards,

Mirza Rustum Baig

Academic Editor

PLOS ONE

Journal Requirements:

2. Please note that in order to use the direct billing option the corresponding author must be affiliated with the chosen institute. Please either amend your manuscript to change the affiliation or corresponding author, or email us at plosone@plos.org with a request to remove this option.

4. Please ensure that you refer to Figures 5 and 6 in your text as, if accepted, production will need this reference to link the reader to the figure.

5. Please upload a copy of Figure 8, to which you refer in your text on page xx. If the figure is no longer to be included as part of the submission please remove all reference to it within the text.

Reviewers' comments:

Reviewer's Responses to Questions

**Comments to the Author**

1. Is the manuscript technically sound, and do the data support the conclusions?

Reviewer #1: Yes

Reviewer #2: Yes

2. Has the statistical analysis been performed appropriately and rigorously? 

Reviewer #1: Yes

Reviewer #2: I Don't Know

3. Have the authors made all data underlying the findings in their manuscript fully available?

Reviewer #1: Yes

Reviewer #2: Yes

4. Is the manuscript presented in an intelligible fashion and written in standard English?

Reviewer #1: Yes

Reviewer #2: Yes

5. Review Comments to the Author

Reviewer #1: I appreciate the authors thought process. However i do have a concerns on the evaluating the trueness of the printing technology with scan bodies. I prefer a standard technique to be used to determine the trueness or exactness of printing technology. The procedure adapted is very generalized toward dentistry. The accuracy should be determined by more definitive methods.

Kindly provide details of sample size estimation, and the method of determining 15 samples.

The writing flow can be improved. It is difficult to comprehend especially in the methodology

Reviewer #2: Dear Authors

Thanks for submitting such interesting research, it was well written, however, I have few comments:

1- Abstract, can you adjust the purpose to include the control/ manufacturer scan bodies

2- Abstract, for the methods, would you please to restructure it and make it more simplified for readers

3- The scan bodies was measured digitally and reverse engineered, that can definitely lead to some deviations between the manufacturer's scan body and the printed one ( not mentioning about the deviations expected of the 3d printing itself), Most of these scan bodies' original STL files are available or even if not you could ask the company to provide you with the files, I used to use B4D and most of the scan bodies are available for exporting, this is just an example.

4- the accuracy of the sprintRay used in this study was 95 microns while for Formlab it was 50 microns how that can affect your results, can you include it into your discussion

5- the authors used Scanning spray, that would affect the accuracy, can you include that as well in your discussion.

6- you have stated "supragingival outer scan body surfaces and excluding areas inside" can you show figure for that, as in my experience with Geomagic control x best fit using one surface in one plane ( just the top of implant platform) will not provide the best fit as expected.

7- for the deviation did you use the RMS or average, please elaborate more about that.

8- figure 4, the abutment-implant assembly blue in color, can elaborate about that, is it the can bodies printed scanned again, not completely clear for me.

best regards

6. PLOS authors have the option to publish the peer review history of their article (what does this mean?). If published, this will include your full peer review and any attached files.

Reviewer #1: No

Reviewer #2: No

---

## [Author Response · Author response to Decision Letter 0]

16 Feb 2023

RESPONSE: The manuscript has been reformatted accordingly.

TEXT CHANGE: Formatting of the manuscript based on PLOS ONE templates.

2. Please note that in order to use the direct billing option the corresponding author must be affiliated with the chosen institute. Please either amend your manuscript to change the affiliation or corresponding author, or email us at plosone@plos.org with a request to remove this option.

RESPONSE: The work was done at Virginia Commonwealth University (VCU). The corresponding author has recently moved to Medical University of South Carolina (MUSC) after the manuscript was submitted. However, the corresponding author still maintains an adjunct faculty appointment at VCU. The affiliations were amended.

TEXT CHANGE: Affiliations have been updated.

RESPONSE: Ethical statement was amended to include the full name of the IRB and the statement of consent waiving.

TEXT CHANGE: The following statement was added.

“The Virginia Commonwealth University Office of Research and Innovation reviewed and approved the study protocol. The use of unidentified data was approved without requiring patient consent (IRB no. HM20009486).”

4. Please ensure that you refer to Figures 5 and 6 in your text as, if accepted, production will need this reference to link the reader to the figure.

RESPONSE: Appreciate the comment.

TEXT CHANGE: All figures are now cited and properly labeled.

5. Please upload a copy of Figure 8, to which you refer in your text on page xx. If the figure is no longer to be included as part of the submission please remove all reference to it within the text.

RESPONSE: There was no Figure 8.

TEXT CHANGE: Citations of figures were amended. Figure 8 citation was removed.

RESPONSE: All references were checked and validated.

TEXT CHANGE: The reference list was checked and updated.

REVIEWER #1

I appreciate the authors thought process. However i do have a concerns on the evaluating the trueness of the printing technology with scan bodies. I prefer a standard technique to be used to determine the trueness or exactness of printing technology. The procedure adapted is very generalized toward dentistry. The accuracy should be determined by more definitive methods.

RESPONSE: Thank you for the kind comments. While we are in agreement with the reviewer that the standard methods should be used, we have adopted the methods to be comparable to clinical measures in implant guide surgery studies. 

TEXT CHANGE: We added the comment below in the Methods.

“The trueness and precision evaluations in this study were determined by the linear and angular deviations of the long axis of the scan body compared to the long axis of the fixture mount instead of standard methods such RMS deviation to be reflective of clinical practice using similar methods as previous implant guided surgery studies and to eliminate minor variation in the reengineering part of the scan body.[23-25]”

Kindly provide details of sample size estimation, and the method of determining 15 samples.

RESPONSE: We appreciate the kind comments. The sample size of 10 for each group was based on a similar previous study (referenece #14)

Donmez MB, Çakmak G, Atalay S, Yilmaz H, Yilmaz B. Trueness and precision of combined healing abutment-scan body system depending on the scan pattern and implant location: An in-vitro study. J Dent. 2022;124: 104169.

TEXT CHANGE: The information on sample size determination was added in the Methods as follows.

“The sample size (n=10) for each group was determined using a similar previous study.[14]”

The writing flow can be improved. It is difficult to comprehend especially in the methodology

RESPONSE: The Methods were revised.

TEXT CHANGE: The Methods were edited, Please see the Methods Section (with and without track changes).

REVIEWER #2

Dear Authors

Thanks for submitting such interesting research, it was well written, however, I have few comments:

1- Abstract, can you adjust the purpose to include the control/ manufacturer scan bodies

RESPONSE: Thank you for pointing this out.

TEXT CHANGE: The purpose in the Abstract is now read.

“This study compared the accuracy of implant scan bodies printed using stereolithography (SLA) and digital light processing (DLP) technologies to the control (manufacturer’s scan body)”

2- Abstract, for the methods, would you please to restructure it and make it more simplified for readers

RESPONSE: We appreciate the comment.

TEXT CHANGE: The methods were restructured to make it more simplified for the read as follows:

“Materials and methods: Scan bodies were printed using SLA (n=10) and DLP (n=10) methods. Ten manufacturer’s scan bodies were used as control. The scan body was placed onto a simulated 3D printed cast with a single implant placed. An implant fixture mount was used as standard. The implant positions were scanned using a laboratory scanner with the fixture mounts, manufacturer’s scan bodies, and the printed scan bodies. The scans of each scan body was then superimposed onto the referenced fixture mount. The 3D angulation and linear deviations were measured.”

3- The scan bodies was measured digitally and reverse engineered, that can definitely lead to some deviations between the manufacturer's scan body and the printed one ( not mentioning about the deviations expected of the 3d printing itself), Most of these scan bodies' original STL files are available or even if not you could ask the company to provide you with the files, I used to use B4D and most of the scan bodies are available for exporting, this is just an example.

RESPONSE: This is a very important comment. We agree with the reviewer that if we have used the original STL file of the scan body, the results would have been closer to the manufacturer’s scan body. However, we want to mimic a situation where a manufacturer’s scan body is not available and if we have to reengineer the scan body.

TEXT CHANGE: The following clarification was added in the Discussion.

“It is important to note that in this study, the scan body was reengineered and printed to simulate a situation when a manufacturer’s scan body is not available such as when the implants are no longer produced by the manufacturer or the implants were placed in another country. The reengineered and printed scan bodies here were aimed to mimic this clinical situation. If the original STL file of the scan body was used, it is possible that the printed scan bodies may provide a closer trueness and precision values as the original scan bodies compared to the reengineered ones.”

4- the accuracy of the sprintRay used in this study was 95 microns while for Formlab it was 50 microns how that can affect your results, can you include it into your discussion

REPONSE: Thank you for pointing this out.

TEXT CHANGE: The printing accuracies of specific printers were added to the Discussion in the paragraph on study’s limitations.

“The accuracy of particular 3D printers within the same technology category may also influence the results. This study utilized SpringRay for the DLP printer and Form 3B for SLA printer. The results, while providing general comparison, may be varied based on differences in particular printer’s printing accuracy.”

5- the authors used Scanning spray, that would affect the accuracy, can you include that as well in your discussion.

RESPONSE: Thank you for pointing this out. We only scan the cast without scanning the scan bodies and the spray was done one time. Then all of the scans for each cast was done in a single round without additional spraying of the cast. 

TEXT CHANGE: The following clarification statement was added into the Methods.

“To minimize the effects of scanning spray to the dimension of the specimen, the spray was performed one time prior to scanning of the fixture mount, and all scan bodies. There was no application of the spray onto the fixture mounts or the scan bodies. All scanning procedures for all four samples (fixture mount, manufacturer’s scan body, SLA scan body, and DLP scan body) in each cast were done consecutively without additional scanning spray application.”

6- you have stated "supragingival outer scan body surfaces and excluding areas inside" can you show figure for that, as in my experience with Geomagic control x best fit using one surface in one plane ( just the top of implant platform) will not provide the best fit as expected.

RESPONSE: We use the four point references as the part of the flat side of the scan body as well as the flat top part of the scan body for the best fit alignment.

TEXT CHANGE: Additional statement was added in the figure legend.

“Once the scans of the cast were aligned using the remaining teeth, the flat part of the scan body with four references points and the flat side of the top of the scan body were then used to define the positioning of the screw access. The central axis of the screw access was then used to compare the linear and angular deviations.”

7- for the deviation did you use the RMS or average, please elaborate more about that.

RESPONSE: We appreciate the comment. We did not use RMS or average deviation but use the long axis of the fixture mount compared to scanbody. 

TEXT CHANGE: A clarification statement was added in the Methods.

“The trueness and precision evaluations in this study were determined by the linear and angular deviations of the long axis of the scan body compared to the long axis of the fixture mount instead of standard methods such RMS deviation to be reflective of clinical practice using similar methods as previous implant guided surgery studies and to eliminate minor variations possible from the reengineering of the scan body.[23-25]”

8- figure 4, the abutment-implant assembly blue in color, can elaborate about that, is it the scan bodies printed scanned again, not completely clear for me.

best regards

RESPONSE: We first aligned the scans of the cast using the teeth and further used the flat part of the scan body, on the side and onto the top, to compare the positional alignment of the screw access. The axis of the screw access was then used to compare the linear and angular deviations.

TEXT CHANGE: Additional clarification statement was added see #6 text change.

---

## [Editor Report · Decision Letter 1]

22 Feb 2023

PONE-D-22-29428R1Accuracy of 3D printed scan bodies for dental implants using two additive manufacturing systems: an in vitro Study

PLOS ONE

Dear Dr. Bencharit,

Thank you for submitting your manuscript to PLOS ONE. After careful consideration, we feel that it has merit but does not fully meet PLOS ONE’s publication criteria as it currently stands. Therefore, we invite you to submit a revised version of the manuscript that addresses the points raised during the review process.

Kindly address the following concerns

Introduction (2nd paragraph, line 5): 

"An accuracy of  a single implant PEEK scanbody is shown on average ~105-127 µm with 0.22°-1.25° in an* in vitro *study.[13] An average range of ~36-57 µm was shown in an in vitro study for a single implant digital impression using a scan body-healing abutment combination system.[14]"

Please state in full for clarity purpose. For example the 'accuracy of a digitized single implant position transfer to a 3D printed cast or digital implant cast using PEEK scanbody'  The surmise in this study is 3D printing scanbodies which are not commercially available in marktet and checking their accuracy of dimensions and features in relation to the original peice. So, the authors also need to explicitly mention in the discussion that in order for this idea to develop in the future they at least need one (1) original scan body (physical) for them to duplicate (scan and create STL file) and 3D print for further use. So, this is a limitation if there is no available scanbody in the market for 3D Printing, unless the STL file can be directly acquired from the manufacturer. Also, the scanbody will vary for implant level and abutment level or tissue level/bone level implants of the same manufacturer. this needs to be emphasized too.   

We look forward to receiving your revised manuscript.

Kind regards,

Mirza Rustum Baig

Academic Editor

PLOS ONE
---

## [Author Response · Author response to Decision Letter 1]

24 Feb 2023

RESPONSES TO REVIEW

COMMENT:

Introduction (2nd paragraph, line 5): 

"An accuracy of a single implant PEEK scanbody is shown on average ~105-127 µm with 0.22°-1.25° in an in vitro study.[13] An average range of ~36-57 µm was shown in an in vitro study for a single implant digital impression using a scan body-healing abutment combination system.[14]"

Please state in full for clarity purpose. For example the 'accuracy of a digitized single implant position transfer to a 3D printed cast or digital implant cast using PEEK scanbody' 

RESPONSE: We appreciate the comment.

TEXT CHANGE: The statements were revised and clarified. 

"An accuracy of a digitized single implant position transfer to a 3D printed cast or digital implant cast using PEEK scanbody of a single implant PEEK scanbody is shown on average ~105-127 µm with 0.22°-1.25° for an in vitro study.[13] Similarly, an average range of ~36-57 µm was shown in an in vitro study for a single implant digital impression using a scan body-healing abutment combination system.[14]"

COMMENT:

The surmise in this study is 3D printing scanbodies which are not commercially available in market and checking their accuracy of dimensions and features in relation to the original piece. So, the authors also need to explicitly mention in the discussion that in order for this idea to develop in the future they at least need one (1) original scan body (physical) for them to duplicate (scan and create STL file) and 3D print for further use. So, this is a limitation if there is no available scanbody in the market for 3D Printing, unless the STL file can be directly acquired from the manufacturer. Also, the scanbody will vary for implant level and abutment level or tissue level/bone level implants of the same manufacturer. this needs to be emphasized too. 

RESPONSE: Thank you very much for this important insight. We added the following clarification in the Discussion in the limitations.

TEXT CHANGE: The limitation statement and future studies statement were amended.

“Finally, this study utilized the known dimension of the manufacturer’s scan body. In real situations when a scan body is not available such as in the older implant systems or implants from other countries, the reverse-engineering process can be challenging and difficult to do. Additionally, variations from implant level or abutment level impression may also further complicate the design and fabrication process of the 3D printed scan body. Considering these limitations, future studies should include improving the design and printing trueness and precision prior to human clinical trials. Reverse-engineering of old dental implant parts as well as scan bodies will be an important topic as part of the future research.”

---

## [Editor Report · Decision Letter 2]

6 Mar 2023

Accuracy of 3D printed scan bodies for dental implants using two additive manufacturing systems: an in vitro Study

PONE-D-22-29428R2

Dear Dr. Bencharit,

We’re pleased to inform you that your manuscript has been judged scientifically suitable for publication and will be formally accepted for publication once it meets all outstanding technical requirements.

Kind regards,

Mirza Rustum Baig

Academic Editor

PLOS ONE

Additional Editor Comments (optional):

Thank you for this important contribution.
---

## [Editor Report · Acceptance letter]

30 Mar 2023

PONE-D-22-29428R2 

Accuracy of 3D printed scan bodies for dental implants using two additive manufacturing systems: an in vitro Study 

Dear Dr. Bencharit:

I'm pleased to inform you that your manuscript has been deemed suitable for publication in PLOS ONE. Congratulations! Your manuscript is now with our production department. 

Kind regards, 

on behalf of

Dr. Mirza Rustum Baig 

Academic Editor

PLOS ONE